# Impact of the Implementation of a Bovine Viral Diarrhea Virus Targeted Vaccine in Dairy Farms: Longitudinal Analysis

**DOI:** 10.3390/vaccines13030319

**Published:** 2025-03-17

**Authors:** Demian Bellido, Diego Wenz, Martin Schang, Facundo Tibaldo Rubiolo, Pablo Mangioni, Emanuel Gumina, Andrés Wigdorovitz, Viviana Parreño

**Affiliations:** 1Vetanco SA, Villa Martelli, Buenos Aires, CP 1603, Argentina; dwenz@vetanco.com (D.W.); mschang@vetanco.com (M.S.); ftibaldo@vetanco.com (F.T.R.); pmangioni@vetanco.com (P.M.); egumina@vetanco.com (E.G.); 2Bioinnovo SA, Villa Martelli, Buenos Aires, CP1603, Argentina; wigdorovitz.andres@inta.gob.ar; 3Incuinta, IVIT INTA, Hurlingham, Buenos Aires, CP1686, Argentina

**Keywords:** bovine viral diarrhea virus, targeted vaccine, dairy farm

## Abstract

Objective: This study evaluated the effect of a novel subunit-targeted vaccine against BVDV in six dairy farms in Argentina. Methods: Reproductive, health, and production parameters—including conception and abortion rates, open days, calves born per insemination, as well as newborn and overall mortality, and milk production—were monitored over a ten-year period (2014–2023). Data were analyzed annually to assess trends and compare the periods before and after vaccine introduction. Results: All parameters showed significant improvement after vaccine incorporation, with an 11% increase in conception rate, a 5% reduction in abortion rate, a 12% increase in calves per insemination rate, and a decrease of 11 open days (8.4%). Additionally, newborn mortality and overall mortality decreased by 33% and 16%, respectively, while milk production increased by 9%. These data were also compared with eight non-vaccinated dairy farms, and significant differences were observed in health and reproductive parameters. Conclusions: These findings indicate that vaccination with an effective non-replicating subunit vaccine can successfully minimize the impact of BVDV in dairy farms.

## 1. Introduction

Bovine Viral Diarrhea Virus (BVDV), considered one of the most important pathogens affecting cattle worldwide, is responsible for enormous production losses in beef and dairy herds [1,2,3]. In this work we studied the impact of the virus in dairy farms and how vaccination with a targeted subunit vaccine can improve production and sanitary indexes. 

BVDV is a member of the genus *Pestivirus* within the family *Flaviviridae* and comprises three species, BVDV-1 (Pestivirus A), BVDV-2 (Pestivirus B), and Hobi-like pestivirus (HoBiPeV; Pestivirus H) [4]. The genome is formed by a single-stranded, positive-sense RNA of 12.3–13 kb encoding a single open reading frame (ORF), which is flanked by 5′ and 3′ untranslated regions (UTRs) [4].

BVDV can affect both the reproductive tract of cows and bulls at different levels, and can infect the semen in artificial insemination practices, leading to low fertility rates or even infertility, embryonic death, fetal loses and abortion, as reviewed by Oguejio [5]. BVDV infection of dams with a non-cytopathic virus biotype during gestation can also lead to persistent fetal infection, if the fetus is infected during the development of the immune system, the calf will not recognize the virus as a pathogen, no immune response will be develop and the animal will become persistently infected, shedding large quantities of virus in its secretions [6,7,8,9]. Moreover, if BVDV infection occurs during the period of fetal organ formation, it results in fetal malformations, most probably due to viral-induced lesions, and to disruption of embryogenesis, leading to the development of congenital defects of several organs, including cerebellar hypoplasia, hydrocephalus, ocular degeneration, thymic hypoplasia, pulmonary hypoplasia, brachygnathism, arthrogryposis, and growth retardation [9,10,11]. These congenital deformities lead to significant reproductive losses in the form of fetal losses, decreased rate of calves born per insemination, decreased availability of replacement heifers in dairy farms, dystocia and cows culled for reproductive problems [5].

Another important consequence of BVDV infection is the induction of immunosuppression associated with the misfunction of immune cells, platelet decrease and leucopenia [12,13,14,15,16]. It also leads to an increase in the chance of superinfections with other pathogens such as *Mannheimia haemolytica*, bovine herpesvirus-1, bovine respiratory syncytial virus, and bovine coronavirus [17,18,19,20,21,22,23], leading to a more severe disease. Bovine respiratory disease syndrome (BRD) may cause the greatest economic impact in the cattle feeding industry (feed lot yards) and dairy farms due to increased morbidity, mortality as well as veterinary-related costs, together with reduced performance [24,25]. Vaccination against BVDV is an important component of prevention and control measures taken in farms as it can prevent clinical signs of BRD, reduce viral spread and reduce the risk of generating new persistently infected (PI) animals. In most countries, only modified live vaccines (MLV) and inactivated vaccines are used. Both vaccine types have historical disadvantages; MLV in terms of safety and inactivated vaccines in terms of immune protection. Over the last decade, our research group has developed and optimized the first targeted subunit BVDV vaccine (BVD-TV). The antigenic part of the BVD-TV is the E2 protein of the virus (Singer strain, BVDV 1a) fused to single chain antibody called APCH, which acts as a targeting molecule [26] and which recognizes an invariant MHC II epitope. This was first developed and tested in swine but has now been shown to cross react with several species, including bovines [27,28,29]. The vaccine is produced in SF9 cells utilizing the baculovirus expression system and was first released to the market in 2018. Vaccine immunogenicity has been tested in guinea pigs and cattle, and efficacy results in feedlot cattle have been previously published [30,31]. In the current work we present new results regarding the improvement of herd reproductive and productive parameters collected during a ten-year period in six commercial dairy farms from the central region of Argentina that incorporated the BVD-TV to their routine vaccination schedules. These vaccinated dairy farms were compared with eight non-vaccinated dairy farms from the same region which act as controls. A total of 346,423 inseminations were studied, leading to 135,773 pregnant dams and 103,771 calves born alive, representing one of the greatest BVDV-vaccine studies ever conducted and reported.

## 2. Materials and Methods

Dairy Farms: The longitudinal study includes six vaccinated dairy farms (VDFs 1 to 6) and eight control dairy farms (CDFs 1 to 8). All VDFs tested positive to BVDV infection. Information of the dairy farms is presented in Appendix A.

Vaccinated dairy farm inclusion criteria were as follows: (i) at least three consecutive years of BVD-TV use; (ii) a minimum of 1000 cows; (iii) use of DairyComp305 software; (iv) no changes in the production system; (v) no implementation of an identification and removal strategy for PI animals, either among newborn calves or recently acquired heifers; and (vi) located in the central region of Argentina, encompassing the provinces of Buenos Aires, Córdoba, Entre Ríos, and Santa Fe.

Control dairy farms inclusion criteria were as follows: (i) no use of the BVD-TV; (ii) use of DairyComp305 software; (iii) no changes in the production system; (iv) no implementation of an identification and removal strategy for PI animals, either among newborn calves or recently acquired heifers; (v) located in the central region of Argentina, encompassing the provinces of Buenos Aires, Córdoba, Entre Ríos, and Santa Fe. 

Vaccines and Vaccine calendars: the different dairy farms, both VDFs and CDFs, have similar routine vaccination protocols:

Respiratory vaccine: this was a combined vaccine containing inactivated cultures of bovine alpha herpesvirus 1 (BoAHV1), bovine viral diarrhea virus (BVDV) and parainfluenza-3 virus (PI3) and bacterins of *Mannheimia haemolytica*, *Pasteurella multocida* and *Histophilus somni*. Calves received the first dose of respiratory vaccine between 45 and 60 days of age and the second dose 21 days later. The third dose was applied when the animals reach 180 kg, around 6 months of age.

Reproductive vaccine: this was a combined vaccine containing inactivated cultures of BoAHV1, BVDV and bacterins of *H somni*, *Campylobacter fetus fetus*, *Campylobacter fetus veneralis*, *Leptospira interrogans* and *Leptospira borgpetersenii*. Heifers received the first dose of reproductive vaccine at the eleventh month of age, the second dose 30 days later and the third dose at pregnancy confirmation. Cows received two doses, the first when they were cleared for breeding (50 to 60 days post-partum) and the second at pregnancy confirmation.

Targeted vaccine against BVDV (BVD-TV): Vedevax Block^®^ (Bioinnovo S.A., Buenos Aires, Argentina) was used as a targeted subunit vaccine against BVDV [30]. Heifers received the first dose of the BVD-TV at the eleventh month of age, the second dose 30 days later and the third 30 days pre-partum. Cows received two doses, the first when they were cleared for breeding (50 to 60 days post-partum) and the second at confirmation of pregnancy (Figure 1). The BDV-TV is incorporated into routine vaccination calendars without replacing or modifying the application schedule of traditional respiratory or reproductive vaccines. Control dairy farms do not receive this vaccine.

Data collection: data from all dairy farms were collected using the DairyComp 305^®^ software (Valley Ag. Software, DairyComp 305 - Version 24.7). Data were collected individually from each dairy farm and then analyzed individually for each dairy farm or as the average of all farms for each parameter evaluated.

Parameters evaluated: in this study all of the reproductive and mortality rates and productive parameters were annually estimated and divided into three stages, as follows: (i) the years before the introduction of the BVD-TV (pre-BVD-TV); (ii) year 0, when the BVD-TV was introduced; and (iii) years after the introduction of the BVD-TV (post-BVD-TV). The different parameters evaluated were as follows:(a)Conception rate: defined as the number of those pregnant out of all the inseminated dams. In the dairy farms included in this study gestation is diagnosed between days 35 and 45 post insemination.(b)Abortion rate: defined as the number of pregnant dams that lost the pregnancy at any gestation point after being diagnosed pregnant.(c)Days open: refers to the interval between calving and conception (DO).(d)Calves per insemination ratio: defined as the ratio between the number of calves born that live more than 24 h and the number of inseminations performed.(e)Neonatal mortality: defined as the death of a live-born calf within the first 24 h of life.(f)Overall mortality: defined as the ratio between the number of death and the average number of animals present in the dairy farm. It does not include neonatal mortality.(g)Milk production: defined as the average milk production per cow per day.


BVDV infection analysis


Viral circulation on the dairy farms was evaluated using a commercial anti-P80 ELISA (CIVTEST BOVIS BVD/BD P80, HIPRA) following the manufacturer’s instructions. Serum samples were taken between February and April 2024. Data from 97 farms (beef and dairy) of the same region collected during 2023 and 2024 were used as controls. In Argentina, where live attenuated vaccines are not permitted, the detection of anti-P80-positive animals indicates viral infection, as inactivated BVDV vaccines and BVD-TV do not induce anti-P80 antibodies due to the non-structural nature of P80.


Statistical analysis:


Regarding the longitudinal studies, on the vaccinated farms pre- and post-vaccination three different statistical analyses were conducted. The first one included the comparison of all the parameter rates two-year pre-vaccination compared with the rate obtained two years post-vaccination, excluding year 0 of vaccine introduction (2YPP). As significant differences were detected, we continue with a second analysis comparing the parameter rates, including the same numbers of years pre- and post-vaccination excluding year 0 (SYPP). Thus, for a farm with five years of vaccination, we compared the parameters obtained post-vaccination with those recorded four years prior to vaccination; for a farm with three years of vaccination, we used the values from two years before vaccination, and so on. The third analysis included all the years available for each dairy farm over a 10-year span from 1 January 2014 to 31 December 2023 (10YS) (Table 1).

In all cases the Chi square test was used for the comparison of proportions. However, when the Chi square test assumptions were not met, Fisher’s exact test was applied to ensure the accuracy of the results. Additionally, a contingency analysis was conducted pre- and post-vaccination, and odds ratios were calculated using the Baptista–Pike method [32].

Mean rates pre- and post-vaccination in each farm and as a whole were compared using a paired Student’s *t* test, using the Welch correction when homoscedasticity assumption was not met. A repeated measures ANOVA was conducted to determine the evolution of each mean parameter through time. The different parameters on the farms were analyzed by a general linear mixed statistical model (GLMM). The model included two main fixed factors: vaccinated farms (VDFs 1 to 6) and time (years pre- and post-vaccination). Animals were included in the model as a random variable. The GLMM analysis was conducted by using the Glimmer function comparisons (lme4 package, R Development Core Team, 2014). Finally, a linear regression analysis was conducted pre- and post-BVD-TV incorporation to evaluate the evolution trend of the indexes.

The longitudinal study to evaluate the evolution trend of the tested parameter in vaccinated (*n* = 6) versus non-vaccinated control farms (n = 8) was conducted by ANCOVA and/or GLMM. The analysis compared the slope of the curves of vaccinated and control farms year by year, during the entire 10-year period from 1 January 2014 to 31 December 2023. Mean rates from VDFs vs CDFs were compared using unpaired Student’s *t* test, using the Welch correction when homoscedasticity assumption was not met.

In all cases we used the GraphPad Prism version 10.0.0 for Mac, GraphPad Software, Boston, MA, USA, www.graphpad.com. Statistical significance was assessed at *p* < 0.05 for all comparisons.

## 3. Results

The viral circulation analysis revealed a significant decrease (Fisher’s exact test, *p* < 0.0001) in measured BVDV circulation and the proportion of animals that were seropositive for antibodies against the P80 viral protein in the dairy farms that included the BVD-TV at the end of the study period compared with the proportion of seropositive cattle in control farms located in the same region (Figure 2). The comparison of P80 antibody prevalences in the farms under study with other farms from the same region tested in 2024 was performed because there were no data available from all of the farms under study in the pre-BVD-TV period.

To analyze the effect of the BVD-TV on the dairy farms, different parameters were evaluated both annually and as a summation of the years before and after the application of the BVD-TV. The parameters can be grouped into the following three categories: (i) reproductive parameters: conception and abortion rate, open days, and calves born per insemination; (ii) mortality rates: neonatal and overall herd mortality; and (iii) productive parameters: milk production.

The results of the comparison of reproductive, mortality rates and productive rates over the same number of years before and after BVD-TV application (SYPP) are shown in Table 2. The SYPP analysis demonstrated a significant improvement in all of the parameters evaluated, which were like those observed in the preliminary 2YPP analysis.

### 3.1. Reproductive Parameters

#### 3.1.1. Conception Rate

The SYPP analysis shows that this index increased significantly, 4.4 points, from 39.1% the years before to 43.5% the years after vaccine introduction. Odds ratio analysis shows that, after the introduction of the BVD-TV, cows have a 22% better chance of becoming pregnant (Table 2).

The 10YS farm-by-farm analysis shows a significant increase in dairy farms 1, 3, 4 and 5 as well as a non-significant increase in farms 2 and 6 (Figure 3 and Appendix A).

The average conception rate of the six VDFs in the 10YS analysis indicated a significant increase of 5.4%, from 37.9% to 43.3% in the post-BTV-TV years (Figure 3a). The overall conception rate shows an increase from 38.2% (51,517/134,915) in the pre-BVD-TV to 44.0% (53,661/122,056) in the post-BVD-TV years.

#### 3.1.2. Abortion Rate

The SYPP analysis shows a significant decrease in the evaluated dairy farms (Table 2). The 10YS farm-by-farm analysis showed a significant decrease in VDFs 1 and 6 as well as a non-significant decrease in VDFs 2, 3, 4 and 5 (Figure 4a,b and Appendix A).

After the introduction of the BVD-TV a significant trend to reduce the abortion rate was observed (Figure 4b). As examples, after 6 year of systematic vaccination farm 1 reduced the abortion rate from 21% to 11% and farm 2 moved from 16% to 7% in three years. The overall result moved from no trend in the years pre-vaccination to a significant linear reduction of 0.77 percentual points per year in the post-vaccination period (R2 = 0.76, *p*-value = 0.02).

#### 3.1.3. Days Open

The SYPP analysis shows a significant decrease of 11 days open, from 131 days open in the years pre-BVD-TV to 120 days open in the years post-BVD-TV (Table 2). The 10YS farm-by-farm analysis showed a significant decrease in VDFs 1, 2, 4 and 6 as well as a non-significant decrease in VDFs 3 and 5 (Figure 5a and Appendix A). The days open average of the six VDFs in the years before and after the application of the BVD-TV indicated a significant decrease of 11 days, from 130 to 119 days (Figure 5b).

#### 3.1.4. Calves Born per Insemination Ratio

The final parameter among the reproductive parameters, encompassing all other reproductive parameters, is the calves born per insemination ratio. The SYPP analysis shows a significant increase of 5%, from 28% in the years pre-BVD-TV to 33% in the years post-BVD-TV in terms of calves born per insemination. OR analysis indicates that the chances of an inseminated cow delivering a live calf were increased by 22% (Table 2). The 10YS analysis shows a significant increase in dairy farms 1, 2 and 4 as well as a non-significant increase in farms 5 and 6. On the other hand, dairy farm 3 showed a non-significant decrease, primarily because in Pre-3, the first year with data from this farm, the index was exceptionally high, at 42% (Figure 6 and Appendix A). The overall mean rates of calves born per insemination of the six dairy farms in the years before and after the application of the BVD-TV indicated a significant increase of 6%, from 27% to 33% (Figure 6b).

The kinetics of reproductive parameters show a clear trend toward improvement after the introduction of the BVD-TV (Figure 7). 

**Figure 7 vaccines-13-00319-f007:**
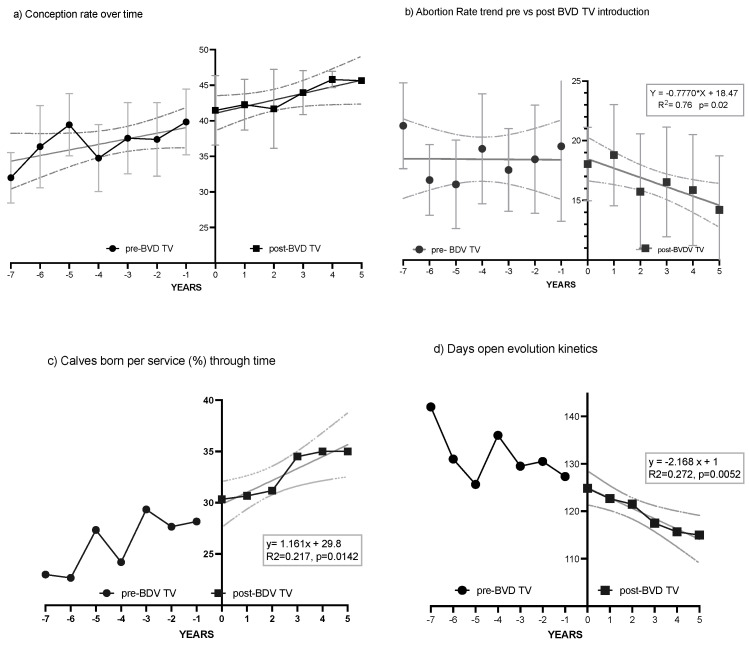
VDF kinetics of the reproductive parameters: Conception rate (**a**), abortion rate (**b**), days open (**c**) and calves per insemination ratio (**d**) pre- and post-BDV-TV introduction. Kinetics of reproductive parameters: (**a**) conception Rate; (**b**) abortion rate; (**c**) days open; (**d**) calves per insemination ratio. In panels (**a**–**d**) the trend of the six VDFs, from years Pre-7 to Post 5 BVD-TV are represented. Year 0 represents the years of introduction of the BVD-TV on the dairy farms. Each point represents the annual mean value of the 6 VDFs, with lines indicating the mean ± standard deviation. **Circles:** pre-BVD-TV. **Squares:** post-BVD-TV. **Dark Gray Lines**: linear regression trend. **Dotted Lines**: IC95 confidence bands.

### 3.2. Mortality Rates

**Neonatal mortality:** The SYPP analysis shows a significant decrease of 1.7%, from 6% during the years pre-BVD-TV to 4.3% days open during the years post-BVD-TV. The OR analysis for neonatal mortality revealed that a calf has a 61% better chance to survive the first 24 h after the application of the BVD-TV against BVDV than before (Table 2). The 10YS farm-by-farm analysis showed a significant decrease in VDFs 1 and 2 as well as a non-significant decrease in VDFs 3 to 6 (Figure 8a and Appendix A). The neonatal mortality average of the six VDFs in the years before and after the application of the BVD-TV presented a significant reduction of 2.8%, from 7% to 4.2% days open (Figure 8b). 

**Overall mortality****:** The SYPP analysis shows a significant decrease of 1.4%, from 9.0% during the years pre-BVD-TV to 7.6% during the years post-BVD-TV. The OR analysis for overall mortality revealed that bovines in the evaluated VDFs have a 26% higher chance to survive after the application of the BVD-TV than before (Table 2). The 10YS farm-by-farm analysis revealed a significant decrease in VDFs 1, 2 and 5 as well as a non-significant decrease in VDFs 4 and 6. On the other hand, dairy farm 3 showed a non-significant increase, primarily influenced by the dry weather that affected the area in the years after the introduction of the vaccine (Figure 8c and Appendix A). The overall mortality average of the six VDFs in the years before and after the application of the BVD-TV presented a significant reduction of 2.6%, from 10.1% to 7.5% (Figure 8d).

### 3.3. Productive Parameters

#### Milk Production

The SYPP analysis shows a significant increase of the mean daily cow production of 2.34 L, from 25.85 L in the years pre-BVD-TV to 28.19 L in the years post-BVD-TV (Table 2). The 10YS farm-by-farm analysis shows a significant increase in dairy farms 1 and 5 as well as a non-significant increase in farms 2, 3, 4 and 6 (Table 3).

The average daily milk production per cow of the six VDFs in the years before and after the application of the BVD-TV presented a significant increase of 2 L, from 25.9 L to 27.9 L pre- vs post-vaccine introduction in the farms (Figure 9a).

### 3.4. Comparison with Control Dairy Farms

The last statistical analysis performed was to compare the trends of the different variables evaluated in the six VDFs, with eight CDFs that did not receive the BVD-TV at any moment over the course of the study (2014–2023). Over the evaluated period (2014–2023), there was a general trend of improvement in dairy farm performance. However, VDFs outperformed CDFs in six of the seven evaluated variables (Figure 10). The conception rate started at the same value in both groups (34%). However, it increased to 42% in VDFs, while in CDFs it improved to 37%. The abortion rate showed a significant decrease in VDFs, dropping by 41.5% over time, from 20% in 2014 to 11.7% in 2023. In contrast, in CDFs there was a decrease from 25% to 23.7%. A similar pattern was observed in the calves per insemination ratio. In VDFs, this index improved from 23% in 2014 to 33% in 2023, representing a 43% increase, whereas in CDFs the rise was smaller, from 29% to 32%. The days open decreased by 18 days in VDFs, from 136 in 2014 to 118 in 2023 (13%), while in CDFs the reduction was only 5 days, from 142 to 137 (3.5%). Regarding mortality, neonatal mortality significantly decreased in VDFs, decreasing from 7.5% in 2014 to 4.3% in 2023, a 43% reduction. In contrast, this index increased in CDFs, reaching 8.6% in 2023. No differences were observed in overall mortality. Finally, daily milk production in VDFs improved by 9.5%, rising from 26.3 L in 2014 to 28.8 L in 2023. In contrast, production in CDFs remained stable throughout the study, starting at 23.6 L and ending at 23.5 L. The linear mix model shows significant differences for the calves per insemination ratio, which encompasses all of the reproductive parameters and one of the health parameters, the neonatal mortality in the VDFs.

## 4. Discussion

BVDV has been eradicated or is in the path to eradication in some European countries due to an extended campaign focused on the detection and elimination of PI animals with or without vaccination [33]. However, in developing countries like Argentina, with large herds and extensive production systems, it is not feasible to expand this control strategy, focused on PI elimination, to the country level. It could be undertaken at the individual farm level, but in the current economic and productive situation it is not feasible to undertake at the regional, province or country levels. Therefore, an alternative approach to the control and minimization of the negative impacts of BVDV infection should be implemented. The strategy to minimize the impact of the virus has been to add a targeted subunit BVD-TV to the current vaccination programs, vaccinating all animals in the herd, when they reach 6 months of age. There is previous work applying a similar strategy that shows the impact of a live-attenuated vaccine (BVDV-MLV) in four dairy farms in Europe [34]. However, in Argentina we cannot use this strategy as MLV are forbidden and only non-replicative vaccines are allowed. Vedevax Block^®^ is the first recombinant targeted subunit vaccine approved for use in bovines worldwide [35]. In the dairy farms under study, combined killed vaccines were also used, before and after the introduction of the recombinant vaccine, with mild effect against BVDV. The synergistic effect of traditional killed vaccines and the BVD-TV has been previously discussed. Although it cannot be completely ruled out, its influence in the improvement of the reproductive and production parameters evaluated appears not to be significant [31].

To measure the impact of the BVD-TV introduction into the vaccine calendar, data from more than 100,000 animals on six farms for a ten-year period were evaluated and compared with eight non-vaccinated dairy farms. Reproductive, mortality rate, and production parameters were tested using different statistical approaches and significant improvements in all of the parameters were observed after vaccine introduction. All farms tested positive for BVDV, which was expected as the virus is endemic in the country [36] and almost 100% of the dairy farms tested positive for infection-related antibodies in milk tanks [37]; however, the average of animals showing signs of infection was significantly reduced in the farms under study after the incorporation of the vaccine, revealing the protective effect of the BVD-TV on the field and the massive impact that this virus may have in the dairy farms.

An important issue is that, in the six farms that we studied, there was, to a greater or lesser extent, an improvement in all seven of the parameters evaluated (conception and abortion rates, days open, calves born per service, neonatal and overall mortality, and average daily milk production) after the introduction of the BVD-TV.

The 6% increment in the rate of calves born per insemination achieved post- versus pre-BVD-TV, from 27% to 33%, resulted in the birth of a total of 2463 more calves post-BVD-TV introduction in the vaccine calendar of the farms. This improvement was achieved together with a reduction of services for 12,861. The improvement in reproductive efficiency has a significant impact on dairy farms, as it affects not only the number of animals but also the milk production and heifer replacements.

The two indexes with the highest economic impact on dairy farm production were as follows: days open and milk production were improved in the VDFs after the introduction of the BVD-TV. In our country, the estimated cost of each additional open day is 6 USD per cow per day [38]. The combination of 18 fewer open days and a 9% increase in daily milk production represents a substantial economic benefit for the vaccinated dairy farms.

To analyze the incidence of factors other than the BVD-TV, such as annual differences due to climate or external factors or a trend toward improved productivity on dairy farms, national data were evaluated from two databases. Different databases presented data in different manners, so we chose the two parameters with year-by-year data to compare: conception rate and milk production.

The conception rate outcome of the dairy farms in the study was compared with the outcome of almost 200 Argentinian dairy farms using the DairyComp 305 software (DC 305 users) (Appendix A). At the national level, the conception rate maintained stability from 2014 to 2020, staying between 31% and 32% before improving to 35% in 2021. This behavior contrasts to that observed in the analyzed dairy farms, where a clear trend to the improvement was observed after the incorporation of the BVD-TV. It is important to note that there was not such a trend before the incorporation of the BVD-TV in these farms. Furthermore, the improvement in reproductive performance places the dairy farms in the top 15% of DC 305 users in Argentina, both in terms of conception and abortion rates.

To analyze milk production, data from the National Minister of Agriculture was used (https://www.magyp.gob.ar/sitio/areas/ss_lecheria/estadisticas/_01_primaria/index.php accessed on 8 July 2024). The data show a peak of production in 2015 and then a decline from 2016 to 2019, a slight recovery from 2020 to 2022 and again a drop in the milk production in 2023. Again, at the national level, milk production does not follow a tendency to improve in the period evaluated as it does in the six dairy farms after the incorporation of the BVD-TV.

Furthermore, in terms of annual or climate factors, all of the years between 2018 and 2021 could have been year 0, pre- or post-vaccination, depending on the farms. This is because this is a longitudinal analysis, the subjects of which did not all begin to participate at the same time. For example, 2020 is year 0 in farm 4, post-BVD-TV in farms 1 to 3 and pre-BVD-TV in farms 5 and 6, so positive or negative annual variances would affect both the pre- and post-vaccination periods depending on the farm.

The results obtained from the six vaccinated dairy farms (VDFs) were compared with those from eight control dairy farms (CDFs) that did not receive the BVD-TV vaccine at any point during the study. All farms were in the same region of the country and had similar management practices. The analysis revealed differences in six of the seven evaluated variables. Moreover, there were significant differences in the calves per insemination ratio, a parameter which encompasses all of the reproductive parameters. Prior to vaccination, the groups were either relatively similar or the vaccinated group exhibited a slightly lower performance. Over the years following vaccination, from 2018 onward, the vaccinated group demonstrated a significant improvement in reproductive efficiency. The same pattern was observed in one of the health parameters analyzed, neonatal mortality.

Considering all of these facts, one cannot discard the influence of external variables in the observed results; however, the influence of such factors would be minor and cannot explain the significant results that were consistently observed in the six different farms that added the BVD-TV to their vaccination program at different time points. Two critical aspects of BVDV infection are immunosuppression and fetal infections, both factors contributing to calf mortality. Interestingly, a recent study reported a reduction in the calves’ mortality index, from 7.45% to 4.38%, after implementing a control program based on the detection and elimination of PI animals [39]. These results are comparable to those obtained in the dairy farms included in this study, which incorporated BVD-TV to their vaccine calendar.

## 5. Conclusions

The addition of a subunit BVD-TV against BVDV into the vaccination calendar of dairy farms contributed to a reduction in the viral circulation and an improvement in productive, reproductive and health parameters such as abortion and conceptions rates, days open, neonatal and general mortality and daily milk production. These findings highlight the importance of incorporating a BVD-TV with an excellent safety profile and of the possibility to differentiate vaccinated from infected animals (DIVA capacity) into comprehensive BVDV control strategies so as to enhance overall herd health and productivity.

## Figures and Tables

**Figure 1 vaccines-13-00319-f001:**
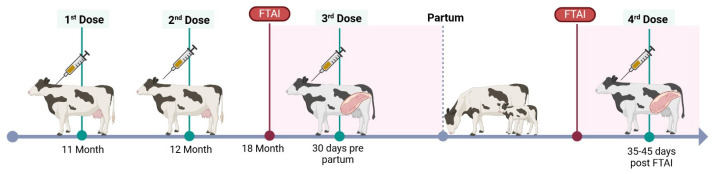
BVD-TV vaccination protocol, FTAI = fixed time artificial insemination.

**Figure 2 vaccines-13-00319-f002:**
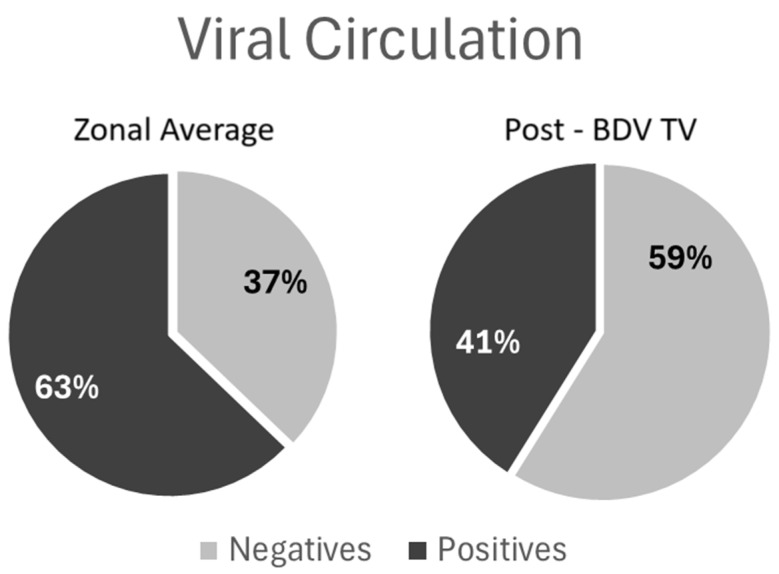
Viral circulation analysis: **Left**: average of 1922 samples from 97 farms, beef and dairy, from the Buenos Aires, Santa Fe and Entre Ríos provinces; **Right**: average of 253 samples from the 6 VDFs at the end of the study period. **Dark Gray**: percentage of seropositive animals for anti-P80 antibodies; **Light Gray**: percentage of seronegative animals for anti-P80 antibodies.

**Figure 3 vaccines-13-00319-f003:**
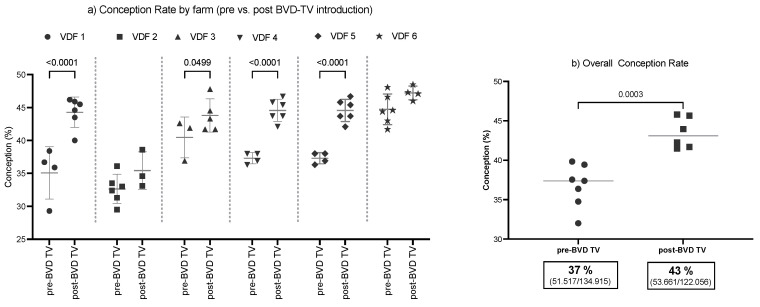
Conception rate evaluation by farm (**a**) and by time (**b**) pre- and post-BDV-TV introduction. (**a**) Conception rate (%) in the six vaccinated dairy farms (VDFs 1–6) before and after the introduction of the BVD-TV vaccine in the immunization calendar. Each point represents the annual conception rate for each farm, with lines indicating the mean ± standard deviation for each group (pre- and post-BVD-TV). Statistically significant differences between groups are indicated (Chi square test, *p* < 0.05). (**b**) Overall conception rate, average per year pre- vs. post-BVD-TV introduction. Each point represents the mean conception rate for all six VDFs in the years pre- and post-BVD-TV introduction from Pre-7 to Post 5. Horizontal lines indicate the mean values for each group (pre- and post-BVD-TV). Statistically significant differences between groups are indicated (Chi square test, *p* < 0.05). The boxes below the graph indicate the total conception rate for each group, along with the total number of pregnant cattle out of the total inseminations performed.

**Figure 4 vaccines-13-00319-f004:**
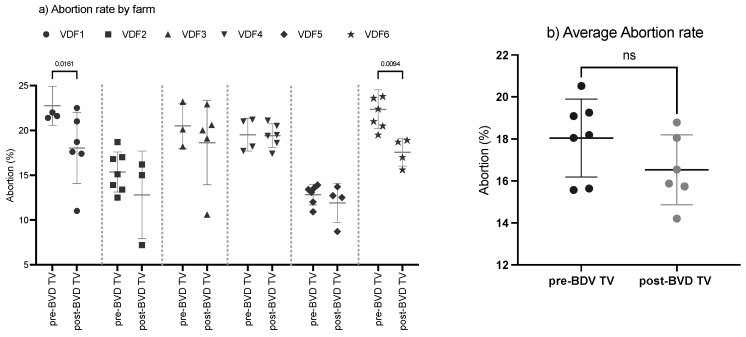
Abortion rate evaluation by farm (**a**) and over time (**b**) pre- and post-BDV-TV introduction. (**a**) Abortion rate (%) in the six vaccinated dairy farms (VDFs 1–6) before and after the introduction of the BVD-TV vaccine in the immunization calendar. Each point represents the annual abortion rate for each farm, with lines indicating the mean ± standard deviation for each group (pre- and post-BVD-TV). Statistically significant differences between groups are indicated. (**b**) Average abortion rate per year, pre- vs. post-BVD-TV introduction in the vaccine calendar. Statistically significant differences between groups are indicated (paired Student’s *t* test).

**Figure 5 vaccines-13-00319-f005:**
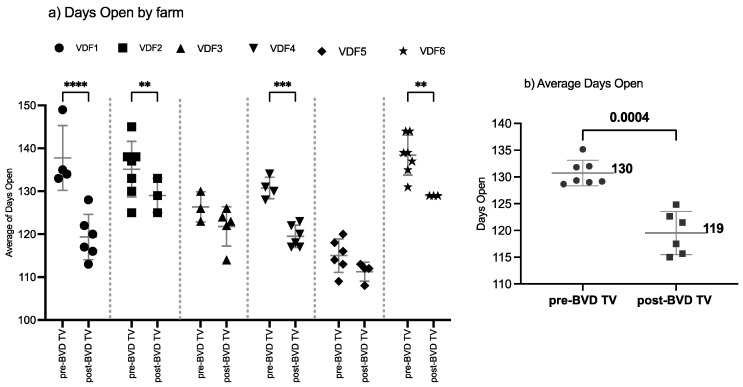
Days open evaluation by farm (**a**) and by year (**b**) pre- and post-BDV-TV introduction. (**a**) Days open in the six vaccinated dairy farms (**VDFs 1**–**6**) before and after the introduction of the BVD-TV vaccine in the immunization calendar. Each point represents the annual days open average for each farm, with lines indicating the mean ± standard deviation for each group (pre- and post-BVD-TV). Statistically significant differences between groups are indicated. (**b**) Overall comparison of the average days open for the six VDFs across time before and after the application of the BVD-TV. Each point represents the annual mean of days open of the six VDFs under study, with lines indicating the overall mean ± standard deviation (pre- and post-BVD-TV). Statistically significant differences between groups are indicated (paired Student’s *t* test). ** < 0.005, *** < 0.0005, and **** < 0.00005.

**Figure 6 vaccines-13-00319-f006:**
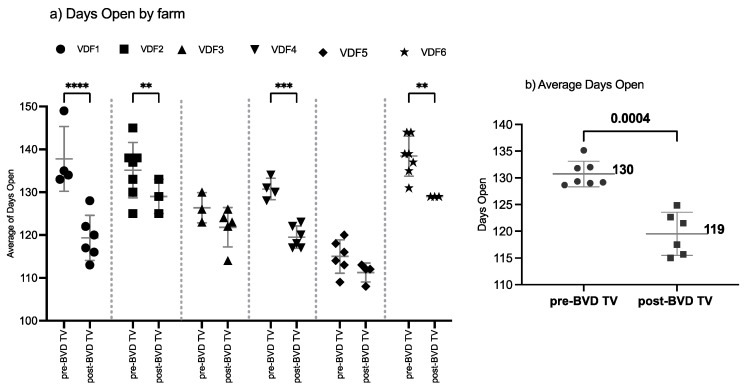
Calves per insemination ratio by farm (**a**) and by year (**b**) pre- and post-BDV-TV introduction. (**a**) Calves per insemination ratio in the six vaccinated dairy farms (VDFs 1–6) before and after the introduction of the BVD-TV in the vaccination calendar. Each point represents the annual rate of calves born per insemination for each farm, with lines indicating the mean ± standard deviation for each group (pre- and post-BVD-TV). Statistically significant differences between groups are indicated (paired Student’s *t* test). (**b**) Average calves per insemination ratio (%) in the six VDFs across the years before and after the introduction of the BVD-TV. Each point represents the annual rate of calves born per insemination of the six VDFs, with lines indicating the mean ± standard deviation. Statistically significant differences between groups are indicated (paired Student’s *t* test). ** < 0.005, *** < 0.0005, and **** < 0.00005.

**Figure 8 vaccines-13-00319-f008:**
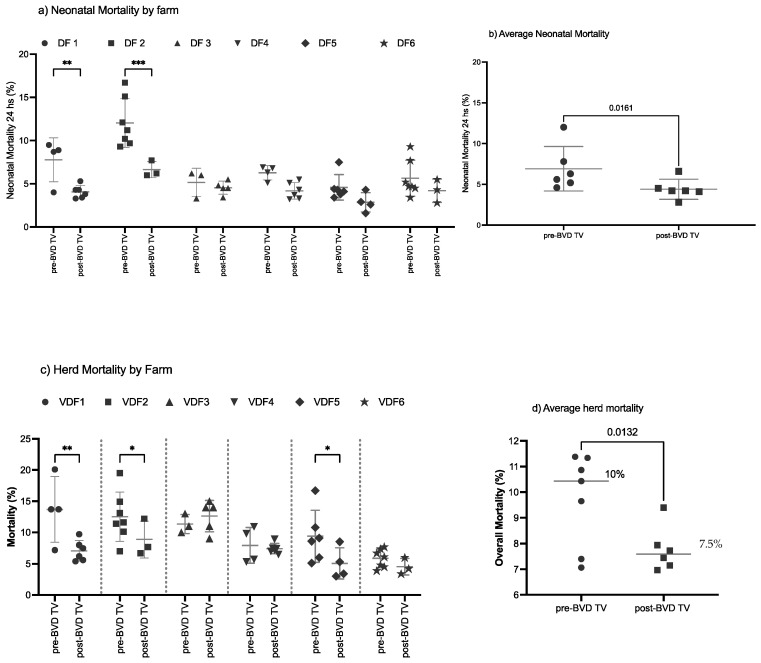
Neonatal and overall mortality, by farm (**a**,**c**) and by year (**b**,**c**) pre- and post-BDV-TV introduction(**a**) Neonatal mortality (%) and (**c**) herd mortality (%) in the six vaccinated dairy farms (**VDFs 1**–**6**) before and after the introduction of the BVD-TV in the immunization calendar. Each point represents the annual mortality (%) of each farm, with lines indicating the mean ± standard deviation for each group (pre- and post-BVD-TV). Statistically significant differences between groups are indicated (paired Student’s *t* test). (**b**) Average neonatal mortality (**b**) and average herd mortality (**d**) pre- vs. post-BVDV-TV introduction. Each point represents the annual mortality of the six VDFs, with lines indicating the mean ± standard deviation. Statistically significant differences between groups are indicated (paired Student’s *t* test). * < 0.05, ** < 0.005, *** < 0.0005.

**Figure 9 vaccines-13-00319-f009:**
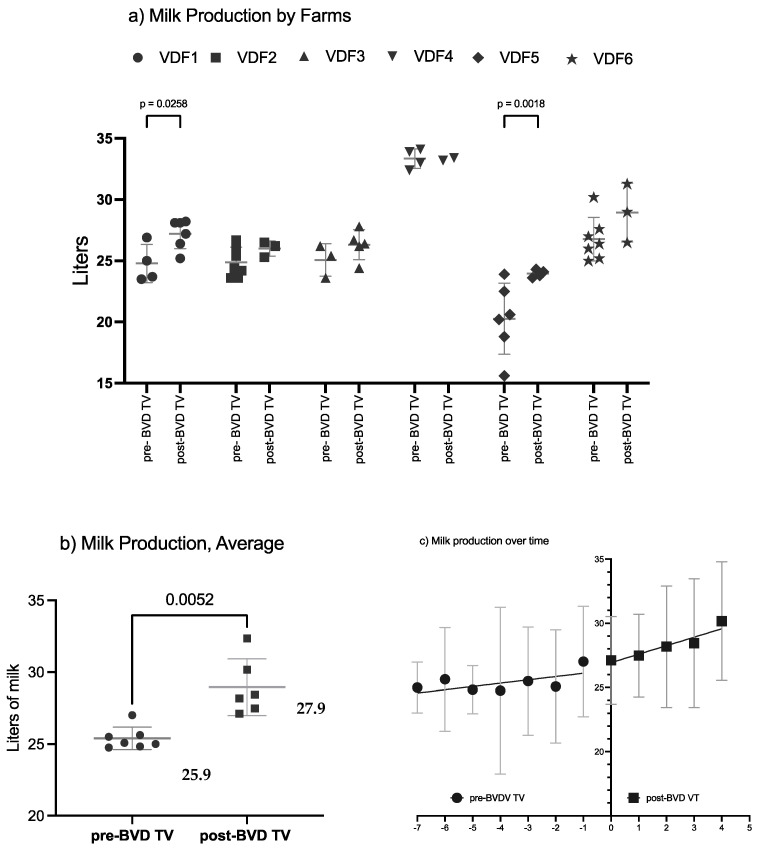
Milk production by farm (**a**), by year (**b**) and by tendency (**c**) pre- and post-BDV-TV introduction. (**a**) Milk production average across the six VDFs before and after the application of the BVD-TV. Each point represents the annual average of daily milk production per cow of the six VDFs with lines indicating the mean ± standard deviation. Statistically significant differences between groups are indicated. (**b**) Average milk production pre- vs. post-BVDV-TV introduction. Each point represents the daily milk production of the six VDFs, with lines indicating the mean ± standard deviation. Statistically significant differences between groups are indicated (paired Student’s *t* test). (**c**) Milk production trend of the six VDFs, from Pre-7 to Post 5 BVD-TV are represented. Year 0 represents the years of introduction of the BVD-TV on the dairy farms. Each point represents the annual mean value of the six VDFs, with lines indicating the mean ± standard deviation. **Circles:** pre-BVD-TV. **Squares:** post-BVD-TV. **Dark Gray Lines**: linear regression trend. **Dot Lines**: IC95 confidence bands.

**Figure 10 vaccines-13-00319-f010:**
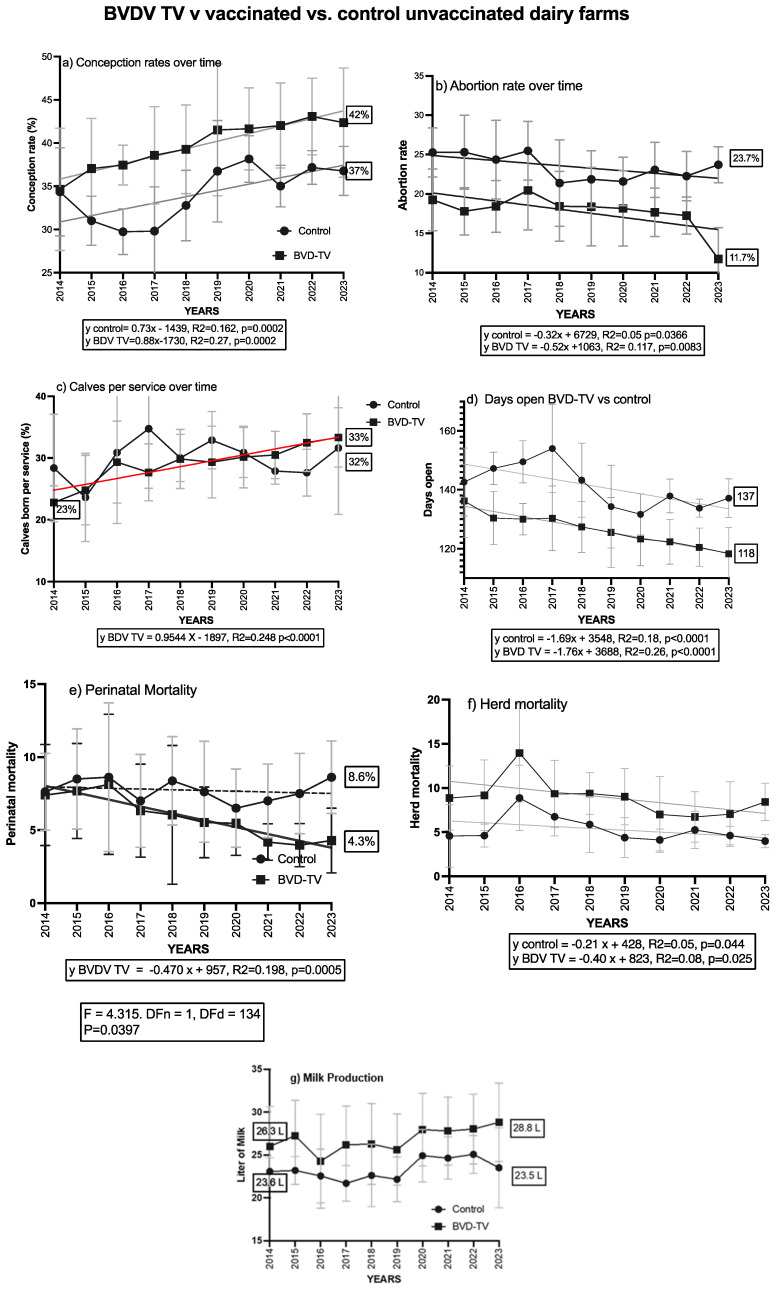
VDFs vs CDFs: trends of all parameters evaluated from 2014 to 2023. Study of parameter evolution through a 10-year period in VDFs vs CDFs. Vaccinated farms introduced the vaccine in 2018 (VDFs x and y, 2019. A linear regression analysis was conducted and the line equation is described when statistically significant (*p* < 0.05)). Only perinatal mortality showed a significant difference in the slopes of the groups.

**Table 1 vaccines-13-00319-t001:** Years studied in each VDF.

Dairy Farm	Pre-7	Pre-6	Pre-5	Pre-4	Pre-3	Pre-2	Pre-1	Year 0	Post 1	Post 2	Post 3	Post 4	Post 5	Total Years
VDF 1				2014	2015	2016	2017	**2018**	2019	2020	2021	2022	2023	10
VDF 2	2014	2015	2016	2017	2018	2019	2020	**2021**	2022	2023				10
VDF 3					2016	2017	2018	**2019**	2020	2021	2022	2023		8
VDF 4				2014	2015	2016	2017	**2018**	2019	2020	2021	2022	2023	10
VDF 5		2014	2015	2016	2017	2018	2019	**2020**	2021	2022	2023			10
VDF 6	2014	2015	2016	2017	2018	2019	2020	**2021**	2022	2023				10

Years studied in each VDF to perform the longitudinal and contingency statistical analyses. Years before the incorporation of the BVD-TV are listed from Pre-1 to Pre-7 and years after the incorporation of the BVD-TV are listed from Post 1 to Post 5. Bold: Year 0, the year of incorporation of the BVD-TV. **Gray**: years used for SYPP analysis. All years, grays and white, were used for the 10YS analysis. Years in the rectangles, Pre-2 and Pre-1 and Post 1 and Post 2, were used for the 2YPP analysis.

**Table 2 vaccines-13-00319-t002:** Parameters evaluated pre and post the introduction of the BVD-TV in six dairy farms.

	Conception Rate (No. of Animals)	Abortion Rate (No. of Animals)	Days Open	Calf Insemination Rate (No. of Animals)	Neonatal Mortality (No. of Animals)	Overall Mortality (No. of Animals)	Milk Production (L/Cow/Day)
Pre-BVD-TV	39.11% (30,612/78,275)	17.84% (5462/30,612)	131	29% (22,543/78,275)	6% (1440/23,983)	8.98%	25.85
Post-BVD-TV	**43.51% *** (36,597/84,108)	**16.83% *** (6161/36,597)	**120 ***	**33% *** (26,971/82,080)	**4% *** (1161/28,132)	**7.57% *** (4866/64,276)	**28.19 ***
Difference	+4.4%	−1%	−11	+4%	−2%	−4%	+2.34
*p*-Value	<0.0001	0.0006	<0.0001	<0.0001	<0.0001	<0.0001	0.0001
OR	1.20	0.93 protective(1.1)	---	1.21	0.67 protective (1.6)	0.89 protective (1.3)	---
IC95	(1.18;1.22)	(0.89;0.97)		(1.18;1.23)	(0.62;0.73)	(0.79;0.86)	

Bold types with asterisks indicate significant differences. **OR**: Odds ratio. **IC95**: 95% confidence interval.

**Table 3 vaccines-13-00319-t003:** Milk production.

	Pre-BVD-TV	Post-BVD-TV	Diff	*p* Value
Dairy Farm 1	**24.8 L**	**27.2** L	**2.4 *** L	**0.0258**
Dairy Farm 2	24.9 L	26.2 L	1.3 L	0.2305
Dairy Farm 3	25.2 L	26.3 L	1.1 L	0.3121
Dairy Farm 4	33.4 L	35.1 L	1.6 L	0.1094
Dairy Farm 5	**20.3** L	**23.9** L	**3.6 *** L	**0.0018**
Dairy Farm 6	26.8 L	28.8 L	2.0 L	0.0779
Average	25.9 L	27.9 L	2.0 L	

Bold type with asterisk indicates significant differences between the means of the pre- vs post-vaccination periods within each farm (paired Student’s *t* test).

## Data Availability

The original data presented in the study are openly available in Appendix A.

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
