# Peer review of "Impact of the Implementation of a Bovine Viral Diarrhea Virus Targeted Vaccine in Dairy Farms: Longitudinal Analysis"

_vaccines, 2025, doi:10.3390/vaccines13030319_

Round 1

Reviewer 1 Report

Comments and Suggestions for Authors

In this paper, the authors attempted to describe the epidemiological survey of immunization by BVDV subunit vaccine in domestic dairy farms. Although it was not clear how the vaccine contributed to dairy productions in each farm, many statistical differences were identified between before and after vaccination. Many farm statuses were analyzed based on longitudinal data, but immunization evidences by BVD-TV were only a few, and restricted and different situation. Therefor the true effect of BVDTV vaccination was unclear. Also in the Discussion, authors estimated these points.

Some other evidence or explanation would be necessary. Also please clarify the following;

1. The title indicates “vaccination program”, in the text, however, program was not estimated. All results were shown the comparison before and after injection of BVD-TV. There was no estimation concerning vaccination program including or not-including BVD-TV. Therefore, the term of “vaccination program” is not appropriate.

2. What is “viral circulation”? As shown in Fig. 3, seropositive rate was decreasing post BVD-TV. Did this mean that vaccination was not effective for uptake of antibodies?

3. In Figs. 5, 9, 11, 16 and 18, there are 7 plots in pre-BVD-TV. The data were based on 6 farms, so plots in post-BVD-TV are 6. What compared? Only Fig. 14, pre and post- are the same number of plots. Moreover, Fig.18 does not correspond with Table 9, where is plot of 33.4 (Farm 4) in pre-BVD-TV in spite of there are seven plots? 35.1 also. Average data in Table 9 and Fig 18 are indicating the same, why and how?

Minor correction

39 citopatic -> cytopathic

48 BDV -> BVD

267 Fig9.a: what is “a:”?

273 Calves -> calves

Reviewer 2 Report

Comments and Suggestions for Authors

This paper describes clinical effectiveness of BVD-TV in dairy farms through longitudinal analysis in the different seven parameters and the findings  are of considerable interest. A few minor revisions are listed bellow.

1. Line 14. PI should be spelled out.

2. Line 37. Delete 2019.

3. Line 39. Cytopathic not citopatic.

4. Lines 4-49. BVDV not BDV-viruses.

5. Line 49. White blood cells.

6. Line 57. BRD should be spelled out.

7. Line 79. Delete respectively.

8. Lines 93-96. Abbreviations should be used.

9. Figure 2 should be Table.

10. Figure 6 lacks some of farm names.

11. Figures 12, 17, and 19 should be improved.

12. Table 9 lacks unit (L).

13. Line 492. DIVA should be spelled out.

14. References. Notation of the journal name should be checked.

Reviewer 3 Report

Comments and Suggestions for Authors

In the discussion, the authors present trends nationally in the parameters of interest from DairyComp 305 data. While useful, I suggest that adding a more select control data set would improve the paper.

 A group of similar size and managed herds from as close geographically as can be matched that did not introduce the BVD-TV vaccine should be evaluated. Getting six dairy farm data sets for controls and using 2017-2020 as pre and 2021-2023 as post would provide a more robust or at least an additional control group. This would enable control for factors such as cyclical trends of BVDV transmission in the region or changes in nutrition, repro technology, weather or factors not considered.

 Mention of herd biosecurity practices is important. Did herds studied and herds in the proposed control group have similar practices particularly as it relates to acquiring bred heifers that are often the source of new BVDV strains and PI calves. Do the herd screen newborns for PI status; did they isolate, and remove those calves? Was that changed in any of the herds? That has that has huge impact on transmission and impacts of virus in vaccinated herds.

The study shows an association but not clear cause and effect. Having the improved comparative control population would strengthen the evidence for BVDV-TV immunity as a causal effect.

 It is unfortunate that pre-virus circulation was not evaluated.

 It was not clear how SYPP was decided and that it was apriori. This is a clear opportunity to manipulate data by altering begin and end dates. It is important that the 2YPP evaluation was done to control for that.

 Table 1 herd size should be defined is that lactating cows, cow numbers or animals on farm at some census date?

 Line 204 it seems a term is missing between when the (and) is applied?

 You should indicate with precision how the dairy farms were selected for inclusion beyond that they had DairyComp 305 data and were willing to vaccinate.

 Description of vaccine protocols before and after were vague. It is not clear, if the killed vaccines were stopped and replaced with the subunit vaccine or if these were added immunizations.  If they were just added then the improvement could be just from another booster. Also, if there were differences in the total vaccine approach between farms it may explain some farm variations.

Round 2

Reviewer 1 Report

Comments and Suggestions for Authors

I think the manuscript is well revised and modified. 

Reviewer 3 Report

Comments and Suggestions for Authors

My concerns were adequately addressed by changes in the manuscript.